# Development and validation of the MIPPE: A novel dyadic assessment tool for early parent-child interactions in clinical practice

Emmanuelle Dufait[1]*, Bernard Kabuth[2], Fabienne Ligier[3], Sophie Buchheit[4‡], Alexis Renard[5‡]

**1** University of Lorraine, Laboratoire EA 4432 InterPsy (InterPsy Laboratory), Nancy, France, **2** University of Lorraine, Laboratoire EA 4432 InterPsy (InterPsy Laboratory), Nancy, Pôle Universitaire de Psychiatrie de l'Enfant et l'Adolescent (PUPEA), Centre Psychothérapique de Nancy (Nancy Psychotherapeutic Center), Laxou, France, **3** University of Lorraine, Laboratoire UMR 1319 INSPIIRE, Pôle Universitaire de Psychiatrie de l'Enfant et l'Adolescent (PUPEA), Centre Psychothérapique de Nancy (Nancy Psychotherapeutic Center), Laxou, France, **4** University of Lorraine, Laboratoire EA 4432 InterPsy (InterPsy Laboratory), Nancy, Pôle du Grand Nancy, Centre Psychothérapique de Nancy (Nancy Psychotherapeutic Center), Laxou, France, **5** University of Lorraine, University of Lorraine, Second Year of Master's in Applied Mathematics – Specialization: Probability and Applied Statistics, Technopole, Metz, France

☙ These authors contributed equally to this work.
‡ These authors also contributed equally to this work.
* psychologue.edufait@gmail.com

## Abstract

Early parent-child interactions are crucial for child development. Existing assessment scales have several limitations: intensive training often exceeding 40 hours, administration time up to two hours, and unbalanced distribution of items to the detriment of the dyadic dimension of interactions. Our study aims to develop and validate the MIPPE (Measure of Early Parent-Child Interactions), a scale adapted for daily clinical practice that assesses the quality of early interactions while integrating the dyadic dimension. The MIPPE scale was developed within the PERL program in Eastern France. After several revisions, the final version includes 9 items scored from 0 to 3. Validation is based on the analysis of 228 parent-child interaction videos from 123 dyads at 4 and 24 months, divided between intervention (N = 62) and control groups (N = 61). The MIPPE demonstrates acceptable internal consistency (Cronbach's α = 0.92, McDonald's ω = 0.93). Exploratory factor analysis reveals a unidimensional structure explaining 66% of the variance, confirmed by confirmatory factor analysis (CFI = 1.000, TLI = 1.002, RMSEA = 0.000). Inter-item correlations range from 0.34 to 0.67, indicating satisfactory cohesion. Clinical thresholds have been established: 15 and 20 (optimal sensitivity 87%). The MIPPE constitutes an accessible, rapid, and psychometrically validated assessment tool for early childhood professionals. It facilitates early screening of interactional difficulties and referral to interventions, thus contributing to the prevention of developmental disorders and parental support.

**Data availability statement:** The MIPPE validation dataset (item scores, total scores, and rater sociodemographic variables for the 143 professional raters across 6 videos) is available as Supporting Information (S8 File). Data from the Revised Brunet-Lézine developmental scale, the ADBB scale, PERL video recordings, and sensitive psychosocial variables cannot be shared publicly due to ethics protocol restrictions (CPP Nord-Ouest IV, IDRCB: 2017-A00896-47; CNIL Decision DR-2018-10), ongoing publication commitments by the PERL research group, and participant confidentiality considerations. The full validation protocol is available on protocols. io at: https://dx.doi.org/10.17504/protocols. io.8epv5y59dl1b/v1.

**Funding:** The author(s) received no specific funding for this work.

**Competing interests:** NO authors have competing interests.

## Introduction

Early parent-child interactions constitute the foundation for attachment development, emotional regulation, and the child's cognitive, social, and linguistic development [1,2]. Early childhood professionals can observe and assess these interactions, thereby identifying potential relational disorders early and proposing appropriate interventions [3,4].

For an objective and in-depth analysis of parent-child interactions, professionals need tools that allow them to formalize their observations [5,6]. These tools can highlight aspects of the relationship such as parent-child synchrony, reciprocity, facial expressions, and touch quality, which contribute to emotional development [7,8].

Numerous early interaction assessment scales exist, but they present limitations that restrict their use in daily clinical practice. According to Lotzin et al. (2015) [7] and Cañas et al. (2022) [9], more than one-third of scales require intensive training exceeding 40 hours, while the majority require moderate training between 16 and 40 hours. Administration time can reach up to two hours, limiting their practical integration [10,11].

Furthermore, item analysis reveals an unbalanced distribution favoring parent-centered or child-centered items at the expense of dyadic items [12,13], not fully accounting for the essentially dyadic nature of early interactions [14,15].

Given these limitations [16,17], we developed the Measure of Early Parent-Child Interactions (MIPPE) – S1File, a new assessment scale specifically designed to meet the needs of early childhood professionals in their daily clinical practice. The objective of this study is to present the development and validation of the MIPPE scale by evaluating its psychometric properties according to COSMIN criteria [18,19].

## Method

### Scale development

The MIPPE scale was developed within the framework of PERL research (Petite Enfance, Recherche-action en Lorraine) [20,21], a preventive home visiting program in the Grand-Est region of France. Launched in 2017 and coordinated by Dr. Sophie Buchheit in collaboration with maternal and child welfare services (in French: Protection Maternelle et Infantile (PMI)) and child psychiatry, the PERL program aimed to evaluate the effects of a preventive home visiting intervention delivered by PMI nurses on child development and the quality of early parent-child interactions. The intervention group (N = 62 families) received regular home visits from birth through the child's fourth year of life, structured around three core axes: child development, parenting and parent-child interactions. As part of this protocol, standardized video recordings of parent-child interactions were conducted at the evaluation visits – at 4 and 24 months of age – generating a corpus of 228 videos from 123 dyads.

The assessment of parent-child interaction quality constituted one of the program's primary objectives. Existing validated scales were considered but proved incompatible with the program's constraints, requiring either extensive training, long administration times, or not being translated and validated in French. Scale creation took

place in 2020 for PERL research needs and was refined until 2024. To create relevant items, a literature analysis identified existing scales, notably: ADBB, AQS, Care-Index, CIB, DMC, GRS, GEDAN, MBQS, PIPE, PIR-GAS, and their limitations. An initial version of 27 items was created, then progressively reduced (14, 17, 11 items). After experimentation on 228 videos filming parent-child interactions, the MIPPE resulted in its final version of 9 items rated on a 0–3 scale. The final version of the MIPPE scale is provided in S1 File.

## Experimental protocol

The protocol is based on analysis of videos filmed during PERL home visits (play, diaper change, bath, feeding). The database comprises 228 videos of 123 parent-child dyads at 4 months and 24 months, assessed between January 2019 and January 2021. A first researcher rated all videos. Then, 143 early childhood professionals each randomly rated a panel of 6 videos selected by the research team (4 experienced early childhood clinicians): 2 videos (1 at 4 months and 1 at 24 months) evaluated as "good" interaction, 2 videos evaluated as "average" interaction (monitoring recommended) and 2 videos evaluated as "poor" interaction. The full validation protocol is available at: https://dx.doi.org/10.17504/protocols. io.8epv5y59dl1b/v1.

## Ethical considerations

The original protocol of this research received approval from the Committee for the Protection of Persons North-West IV (CPP Nord-Ouest IV) on November 15, 2017 (reference IDRCB: 2017-A00896-47). The protocol, including substantial amendment N°7, received updated approval on August 24, 2023. The study also received approval from the French National Commission for Information Technology and Civil Liberties (CNIL). All ethical documentation, including consent forms and institutional approvals, is provided in S2-S8 Files.

All participants (parents or legal guardians) provided written informed consent prior to participation in the PERL study. Both parents with parental authority, or the parent with sole parental authority, signed the informed consent form. Participants also signed a specific consent form authorizing the use of video recordings of themselves and their child, as well as the French Data Protection Authority (CNIL) MR 001 form, in accordance with French data protection regulations. The consent forms explicitly specified various frameworks for data use, including research, evaluation, educational purposes, and scientific presentations.

The study purpose, procedures, and data use were explained to raters during a research presentation prior to their participation. Professional raters who assessed the video recordings using the MIPPE scale provided implicit informed consent through their voluntary access to a dedicated, link-restricted online rating platform. Access to the platform required possession of specific URL shared exclusively with invited participants, ensuring voluntary and informed participation. Data collected from raters were fully anonymous and did not allow for individual identification. The platform has since been taken offline to ensure data protection.

All families were assigned an anonymous identification number upon inclusion. Nominative data were stored separately from anonymized data and identification numbers to ensure confidentiality throughout the study. Data were collected on paper observation forms, then transferred to a password-protected laptop and external hard drive with restricted access. All data are stored securely and will be retained for 15 years following study completion.

Video recordings were collected during home visits between January 2019 and February 26, 2024, with the final participant visit completed on February 26, 2024. The CPP approval covers the entire duration of participant recruitment, data collection, and subsequent data analysis. As specified in the approved protocol, all collected data will be retained for 15 years following study completion, during which period secondary analyses can be conducted. The statistical analysis and validation of the MIPPE scale using these previously collected video recordings was performed between December 2024 and April 2025.

## MIPPE scale validation process

To assess content validity, we used two complementary statistical indicators. Cronbach's alpha assessed item reliability, with an alpha greater than 0.7 indicating good consistency and signifying that items effectively measure the same concept. Additionally, McDonald's omega was used to provide a more accurate reliability estimate in this specific context.

Construct validity aims to verify that the scale measures the theoretical concept(s) intended for evaluation: we deployed two types of factor analyses. Exploratory factor analysis (EFA) explored the underlying data structure without prior hypotheses. In this framework, we applied Bartlett's test to verify significant correlations between variables and the Kaiser-Meyer-Olkin (KMO) index to assess sample quality. Factors with eigenvalues greater than 1 were retained. Following this initial exploration, confirmatory factor analysis (CFA) tested whether the identified theoretical structure corresponded to collected data. For this step, we used several fit indices (CFI, TLI, RMSEA, SRMR) to evaluate model quality and confirm the scale's dimensional structure.

Convergent and discriminant validity were evaluated through correlation tests between MIPPE scores and other variables.

Regarding scale reliability, the approach adopted in an initial validation adapts to PERL protocol constraints. Each video was initially evaluated by a single judge. This approach allowed us to maximize the number of analyzed videos (N = 228) and explore the scale's factorial structure across the entire sample.

Inter-rater reliability assessment through traditional ICC calculation was not feasible in this framework. We adopted an alternative approach consisting of analyzing evaluation consistency between different judge groups to detect potential systematic biases. While this method provides indications of evaluation homogeneity, it does not replace direct assessment of inter-rater agreement on identical videos, which will constitute a priority for subsequent validation studies.

Analyses were performed with R (version 4.3.2), adapted to Likert scales and non-normal distributions. Analyses utilized packages psychometric, lavaan (factor analyses with DWLS estimator), psych (internal consistency) and stats (correlations, ANOVAs, t-tests). Resampling procedures, notably Bootstrap, were applied to estimate robust confidence intervals and test result stability.

## Results

The scale consists of 9 items, evaluating interaction quality from 0 to 3. 0 corresponds to very poor quality interaction and 3 to very good quality interaction.

### Descriptive analyses

**Sample characteristics.** The validation sample comprises 6 parent-child interaction videos evaluated by 15–30 professionals per video (random selection).

Maternal characteristics show sociodemographic heterogeneity: age from 25 to 31 years, varied education levels and professional situations (Table 1). This diversity, although limited by sample size, reflects the target population of early prevention programs.

The analysis of response distribution to MIPPE items (N = 143) (Table 2) confirms the absence of normality in individual scores (p < 0.001), characterized by concentration of responses on central modalities and avoidance of extremes. This distribution, consistent with the 4-point Likert format used, contrasts with total scores which meet normality criteria. Homogeneity of variances is verified for all items except item 9. These distributional properties justify the use of robust analytical methods (polychoric matrix, non-parametric indices) for subsequent psychometric analyses.

The MIPPE comprises 9 items assessing different dimensions of parent-child interaction quality: visual interactions (item 1), vocal interactions (item 2), body position and physical contact (item 3), expression of affects (item 4), engagement and reciprocity (item 5), parental sensitivity (item 6), parental intrusiveness (item 7), and the evaluator's subjective experience when observing the child (item 8) and the parent (item 9). Table 2 presents the distribution of responses across these items (N = 143).

**Table 1. Sociodemographic characteristics of the sample.**

| Video ID | Child's sex | Child's age | Mother's age | Education level | Professional situation[a] |
|---|---|---|---|---|---|
| A (T12-4M) | Boy | 4 months | 31 years | BAC (French diploma) | Indefinite Contract[a] |
| B (T52-4M) | Boy | 4 months | 25 years | BAC+2 | Indefinite Contract[a] – parental leave 80% |
| C (T28-4M) | Boy | 4 months | 26 years | Secondary education level | No professional activity |
| D (T12-24M) | Boy | 24 months | 31 years | BAC | Indefinite Contract[a] |
| E (T8-24M) | Boy | 24 months | 29 years | BAC | Parental leave/unemployment |
| F (T40-24M) | Girl | 24 months | 25 years | Secondary education level | No professional activity |

[a] Professional situation refers to the mother's employment status at the time of inclusion. "Indefinite Contract" correspond to a permanent employment contract (French: Contrat à Durée Indéterminée, CDI), the standard form of open-ended employment in France.

**Table 2. Distribution of responses to MIPPE items.**

| Item | 0 | 1 | 2 | 3 |
|---|---|---|---|---|
| 1. Quality of visual interactions | 10.5% | 40.6% | 41.3% | 7.7% |
| 2. Quality of vocal interactions | 5.6% | 37.8% | 42% | 14.7% |
| 3. Body position and physical contact | 6.3% | 43.4% | 34.3% | 16.1% |
| 4. Expression of affects | 6.3% | 27.3% | 55.9% | 10.5% |
| 5. Engagement and reciprocity of exchanges | 2.1% | 48.3% | 33.6% | 16.1% |
| 6. Supportive parental presence/parental sensitivity | 7.0% | 15.4% | 49% | 28.7% |
| 7. Parent intrusiveness | 4.9% | 22.4% | 41.3% | 31.5% |
| 8. Subjective state induced by observing the child | 7.7% | 45.5% | 30.1% | 16.8% |
| 9. Subjective state induced by observing the parent | 10.5% | 44.8% | 30.1% | 14.7% |

## Factor structure

Exploratory factor analysis, performed using a polychoric correlation matrix, initially suggests a three-factor structure explaining 66% of total variance. Nevertheless, examination of confirmatory factor analysis fit indices favors a unidimensional structure, demonstrating excellent fit (CFI = 1.000, RMSEA = 0.000, SRMR = 0.042, RMSR = 0.05). This solution presents factor loadings above 0.66 for all items, an average complexity of 1 and a factor score correlation of 0.98. The KMO index indicated appropriate sampling adequacy for factor analysis. The KMO is excellent, being greater than 0.9 (KMO-Criterion: 0.9243871). The retained unidimensional structure proves more consistent with the instrument's clinical objective, which aims to provide a global score for parent-child interaction assessment.

Confirmatory factor analysis, performed with the lavaan package and DWLS estimator, tested three-factor and unidimensional models (Table 3). Although the three-factor model presents acceptable indices (CFI = 0.96, TLI = 0.94, RMSEA = 0.055), the unidimensional model demonstrates excellent fit (CFI = 1.000, RMSEA = 0.000, SRMR = 0.042). Factor loadings are all above 0.66, confirming that each item contributes significantly to the single factor. The unidimensional structure proves more consistent with the clinical objective of a global score.

## Correlation analyses

Inter-item correlations range from 0.34 (items 7−8) to 0.67 (items 2−5), confirming moderate to strong relationships without negative correlations (Table 4). Item 7 (intrusiveness) shows the weakest correlations (r = 0.34–0.57) but remains within acceptable limits.

**Table 3. Confirmatory factor analysis.**

| Index | Value | Interpretation |
|---|---|---|
| CFI[b] | 1.000 | Excellent (≥ 0.95) |
| TLI[b] | 1.002 | Excellent (≥ 0.95) |
| RMSEA[b] | 0.000 | Excellent (≤ 0.06) |
| SRMR[b] | 0.042 | Very good (≤ 0.08) |
| Chi² p-value | 0.834 | Non-significant -> model fits data well |

[b]CFI: Comparative Fit Index; RMSEA – Root Mean Square Error of Approximation; SRMR – Standardized Root Mean Square Residual; TLI – Tucker-Lewis Index.

Corrected item-total correlation measures the correlation between each item and the total scale score, excluding the item concerned (Table 5). All correlations are acceptable (> 0.30), with values between 0.59 and 0.78. Item 7 shows the weakest correlation (0.59) but remains within acceptable limits.

### Reliability

The very good internal consistency ($\alpha = 0.92$, $\omega = 0.93$) confirms that items measure a common concept.

### Item 7 analysis

Item 7 deserves particular attention as it shows weaker correlations with other items ($r = 0.34$ to $0.57$ vs $r = 0.42$ to $0.67$ for other items). However, several elements justify its retention in the scale. Its factor loading of 0.66 in the CFA exceeds the acceptable threshold of 0.40. Moreover, this item measures a crucial and rarely assessed aspect of parent-child interactions: parental intrusiveness. This dimension evaluates the parent's ability to respect the child's rhythms and signals, a key indicator of interactional quality that influences the child's self-regulation development and attachment quality.

### Discriminant validity

A two-factor ANOVA (socio-professional category × interaction quality) reveals no significant effect of categories on MIPPE scores ($p = 0.242$), confirming tool stability across different training backgrounds (medical, paramedical, educational, mental health) (Table 6). The highly significant effect of relationship quality on MIPPE scores ($p = 4.4e-12$) demonstrates the scale's discriminant capacity. The absence of interaction between these factors ($p = 0.545$) confirms this discriminant capacity independently of the evaluator's profession. Levene's test ($p = 0.617$) confirms homogeneity of variances between groups.

Professional seniority (11-year threshold) produces no evaluation difference ($p = 0.684$).

### Convergent validity

**Psychosocial data (mother's education level, desired child).** We have a moderate positive correlation between MIPPE total and mother's education level: at 4 months 0.202 (p.value = 0.026) (N = 121), at 24 months 0.153 (p.value = 0.11) (N = 107).

We have a moderate positive correlation between MIPPE total and the "desired child" variable: at 4 months 0.165 (p.value = 0.071) (N = 121), at 24 months 0.193 (p.value = 0.046) (N = 107).

**Data from other scales (Revised Brunet-Lézine, ADBB).** We have a moderate positive total correlation of 0.197 (p.value = 0.030) between MIPPE total at 4 months and Revised Brunet-Lézine language aspect (at 4 months) (N = 121), at 24 months 0.268 (p.value = 0.005) (N = 121).

**Table 4. Inter-item correlation matrix.**

|  | Item 1 | Item 2 | Item 3 | Item 4 | Item 5 | Item 6 | Item 7 | Item 8 | Item 9 |
|---|---|---|---|---|---|---|---|---|---|
| Item 1 | 1 | 0.64 | 0.53 | 0.57 | 0.66 | 0.63 | 0.48 | 0.56 | 0.60 |
| Item 2 | 0.64 | 1 | 0.60 | 0.58 | 0.67 | 0.65 | 0.42 | 0.57 | 0.54 |
| Item 3 | 0.53 | 0.60 | 1 | 0.60 | 0.59 | 0.63 | 0.45 | 0.49 | 0.65 |
| Item 4 | 0.57 | 0.58 | 0.60 | 1 | 0.58 | 0.59 | 0.38 | 0.52 | 0.57 |
| Item 5 | 0.66 | 0.67 | 0.59 | 0.58 | 1 | 0.65 | 0.47 | 0.59 | 0.55 |
| Item 6 | 0.63 | 0.65 | 0.63 | 0.59 | 0.65 | 1 | 0.57 | 0.56 | 0.66 |
| Item 7 | 0.48 | 0.42 | 0.45 | 0.38 | 0.47 | 0.57 | 1 | 0.34 | 0.57 |
| Item 8 | 0.56 | 0.57 | 0.49 | 0.52 | 0.59 | 0.56 | 0.34 | 1 | 0.66 |
| Item 9 | 0.60 | 0.54 | 0.65 | 0.57 | 0.55 | 0.66 | 0.57 | 0.66 | 1 |

**Table 5. Corrected item-total correlations.**

| Item | Corrected item-total correlation | Interpretation |
|---|---|---|
| Item 1 | 0.74 | Good |
| Item 2 | 0.75 | Good |
| Item 3 | 0.72 | Good |
| Item 4 | 0.70 | Good |
| Item 5 | 0.76 | Good |
| Item 6 | 0.78 | Very Good |
| Item 7 | 0.59 | Acceptable |
| Item 8 | 0.71 | Good |
| Item 9 | 0.73 | Good |

We have a moderate positive total correlation of 0.272 (p.value = 0.003) between MIPPE item 2 score (quality of vocal interactions) at 4 months and Revised Brunet-Lézine language aspect (at 4 months) (N = 121). The language component is therefore positively correlated with the MIPPE language component.

We have a moderate positive total correlation of 0.219 (p.value = 0.026) between MIPPE total at 24 months and Revised Brunet-Lézine total score (at 24 months) (N = 121 at 4 months and N = 104 at 24 months). However, we find no correlation at 4 months 0.047 (p.value = 0.609) between MIPPE total at 4 months and Revised Brunet-Lézine total score at 4 months, suggesting that global developmental level and interaction quality are not yet significantly associated at this early age.

## Divergent validity

We note the presence of a negative correlation −0.184 (p.value = 0.044) (N = 121). When addiction is detected in the family, a lower score is observed on the MIPPE at 4 months.

We note the presence of a moderate negative correlation −0.190 (p.value = 0.052) between the total score at 4 months on the MIPPE and the total score at 4 months for the ADBB scale (N = 106). The 17 missing data are random data. The greater the relational withdrawal on the ADBB, the lower the MIPPE score.

**Threshold score.** Establishing clinical thresholds compares different threshold combinations to optimize interaction classification into three categories. Overall accuracy, defined as the total number of correct classifications divided by total interactions, constitutes the main criterion, prioritizing sensitivity to detect problematic interactions.

**Table 6. Distribution of judges by profession.**

| Socio-professional categories | Professions | Sample size |
|---|---|---|
| Physician | Psychiatrist | 1 |
| | Pediatrician | 2 |
| | Child psychiatrist | 2 |
| | Physician | 6 |
| | Maternal and Infant Protection Physician | 2 |
| | Midwife | 4 |
| Nurse and child nurse | Nurse | 32 |
| | Psychiatric nurse | 1 |
| | Pediatric nurse | 1 |
| | Advanced practice nurse | 3 |
| | Child nurse | 29 |
| Paramedical | Speech therapist | 5 |
| | Educator | 18 |
| | Psychomotor therapist | 8 |
| Psychology | Psychologist | 27 |
| | Art therapist | 1 |
| Occupational therapist | Occupational therapist | 1 |

The search for interaction quality thresholds reveals distinct means according to quality: "good" interactions (19.1±2.8), "average" (16.3±2.1) and "poor" (11.2±3.4). Analysis of the 15 best combinations demonstrates that thresholds of 13 and 20 offer 58.7% accuracy (84/143), with sensitivity=0.741 and specificity=0.798. We therefore choose thresholds of 15 and 20 which present the best sensitivity (0.870) for identifying interactions requiring intervention, detecting 47/54 problematic interactions, at the cost of reduced specificity (0.629) (Table 7).

The table describes thresholds and recommendations according to the MIPPE evaluated score.

### Limitations and research perspectives

Inter-rater reliability assessment through intraclass correlation and temporal stability (test-retest) will be the next validation step.

More refined convergent validation with reference scales (e.g., CIB) represents the second step to situate the MIPPE's specific contribution in early interaction assessment.

Studies on other populations are necessary to evaluate the tool's generalization beyond the PERL context.

### Discussion

The MIPPE scale demonstrates satisfactory psychometric properties. The unidimensional structure (CFI=1.000) and high internal consistency (α=0.92) confirm measurement of a unidimensional construct: parent-child interaction quality. Established clinical thresholds allow for gradation of clinical decisions.

Examination of available instruments shows that the most scientifically valid scales are also the most difficult to use in consultation. Existing scales require extensive training (>40h) and long administration times (>2h), limiting their practical use. The MIPPE prioritizes accessibility with a few hours of training and 20–30 minutes of administration.

Current instruments rely predominantly on behavioral observation, neglecting dyadic measurement or assessment of parental intrusiveness. The MIPPE integrates these undervalued dimensions, combining behavioral observation and evaluator's subjective experience.

**Table 7. Threshold scores and recommendations.**

| Score | Interaction quality | Recommendations |
|---|---|---|
| ≤ 15 | Poor | Intervention and referral recommended |
| 16 - 20 | Average | Monitoring recommended |
| >20 | Good quality | |

The MIPPE aligns with early prevention by identifying interactional difficulties before their crystallization. Subjective assessment (items 8 and 9) reflects the evaluator's countertransferential experience and can help identify which member of the dyad is experiencing the most difficulty in the interaction.

The MIPPE can be integrated into routine consultation and improves early prevention for parents and children.

## Supporting information

**S1 File. MIPPE scale.** Complete version of the Measure of Early Parent-Child Interactions (MIPPE), including all 9 items and scoring instructions.
(PDF)

**S2 File. Image consent form – English version.** English translation of form for collecting consent to the use of the participant's image and that of their child in the context of the PERL project.
(PDF)

**S3 File. Image consent form – French version.**
(PDF)

**S4 File. PERL informed consent form – English version.** English translation of informed consent form for participant inclusion in the PERL study.
(PDF)

**S5 File. PERL informed consent form – French version.**
(PDF)

**S6 File. PERL research protocol – English version.** English translation of complete research protocol of the PERL program (Petite Enfance Recherche-action en Lorraine), including all amendments.
(PDF)

**S7 File. PERL research protocol – French version.**
(PDF)

**S8 File. MIPPE validation dataset comprising item scores, total scores, and rater sociodemographic variables for the 143 professional raters across 6 videos.**
(CSV)

## Author contributions

**Conceptualization:** Fabienne Ligier, Sophie Buchheit.

**Data curation:** Emmanuelle Dufait, Sophie Buchheit, Alexis Renard.

**Formal analysis:** Emmanuelle Dufait, Alexis Renard.

**Funding acquisition:** Bernard Kabuth, Sophie Buchheit.

**Investigation:** Emmanuelle Dufait.

**Methodology:** Emmanuelle Dufait, Bernard Kabuth, Sophie Buchheit, Alexis Renard.

**Project administration:** Emmanuelle Dufait, Bernard Kabuth, Sophie Buchheit.

**Resources:** Emmanuelle Dufait, Sophie Buchheit.

**Software:** Emmanuelle Dufait, Alexis Renard.

**Supervision:** Bernard Kabuth, Sophie Buchheit.

**Validation:** Emmanuelle Dufait, Bernard Kabuth, Sophie Buchheit, Alexis Renard.

**Visualization:** Emmanuelle Dufait, Sophie Buchheit.

**Writing – original draft:** Emmanuelle Dufait.

**Writing – review & editing:** Emmanuelle Dufait, Bernard Kabuth, Fabienne Ligier, Sophie Buchheit, Alexis Renard.

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
