## [Decision Letter · Decision Letter 0]

18 Mar 2026

Dear Dr. Dufait,

Thank you for submitting your manuscript to PLOS ONE. After careful consideration, we feel that it has merit but does not fully meet PLOS ONE’s publication criteria as it currently stands. Therefore, we invite you to submit a revised version of the manuscript that addresses the points raised during the review process.

We look forward to receiving your revised manuscript.

Kind regards,

Randall Waechter

Academic Editor

PLOS One

Journal Requirements:

2. Please note that your Data Availability Statement is currently missing the repository name and/or the DOI/accession number of each dataset OR a direct link to access each database. If your manuscript is accepted for publication, you will be asked to provide these details on a very short timeline. We therefore suggest that you provide this information now, though we will not hold up the peer review process if you are unable.

5. Please include captions for your Supporting Information files at the end of your manuscript, and update any in-text citations to match accordingly. Please see our Supporting Information guidelines for more information: http://journals.plos.org/plosone/s/supporting-information....

6. We are unable to open your Supporting Information file “consentement0510.docx and renamed_a3a62.pdf” Please kindly revise as necessary and re-upload.

Additional Editor Comments:

Thank you for submitting this manuscript. There are reviewer comments to address before proceeding with publication. Please address each of the comments and resubmit.

Reviewer's Responses to Questions

**Comments to the Author**

1. Is the manuscript technically sound, and do the data support the conclusions?

Reviewer #1: Yes

2. Has the statistical analysis been performed appropriately and rigorously?

Reviewer #1: Yes

3. Have the authors made all data underlying the findings in their manuscript fully available?

Reviewer #1: Yes

4. Is the manuscript presented in an intelligible fashion and written in standard English?

Reviewer #1: Yes

Reviewer #1: Identify KMO in the first instance.

Consider providing a bit more context, i.e. how the measure aligns with the protocol, how the protocol came about.

Explain "professional situation" in Table 1

Explain Items 8 and 9 in Table 2

Table 6 is impressive and adds to your scale's credibility signifcantly

Although there are numerous supporting documents I did not find Appendix 1, the scale itself, which may help to explain items 8 and 9, and perhaps also the parental intrusiveness item. Please verify that the Appendix is attached.

.

Reviewer #1: **Yes:** Barbara LandonBarbara LandonBarbara LandonBarbara Landon

---

## [Author Response · Author response to Decision Letter 1]

30 Mar 2026

We thank the Academic Editor (Dr. Randall Waechter) and Reviewer #1 (Dr. Barbara Landon) for their thorough and constructive feedback. All comments have been addressed in full in the Response to Reviewers document. The main revisions are summarized below:

Journal Requirements: All PLOS ONE formatting requirements have been addressed: file naming conventions updated (S1–S12 Files), a Data Availability Statement added, ethics statement clarified for both participant groups, captions added for all Supporting Information files, corrupted file replaced, and the reference list fully revised in Vancouver format with DOIs. The COSMIN criteria are now supported by the two appropriate Mokkink et al. (2010) references. The MIPPE validation dataset is provided as S12 File. A partial data sharing exemption is requested for PERL video recordings, Brunet-Lézine, ADBB, and sensitive psychosocial variables, with full justification provided in the Response to Reviewers and Data Availability Statement. The full validation protocol has been deposited on protocols.io (DOI: https://dx.doi.org/10.17504/protocols.io.8epv5y59dl1b/v1).

Reviewer #1 — Dr. Barbara Landon: (1) KMO is now defined in full at first mention. (2) The Scale Development section has been substantially expanded to describe the origins of the PERL program and the role of Professor Antoine Guédeney in guiding scale development. (3) A footnote has been added to Table 1 explaining "professional situation" and "Indefinite Contract" (CDI). (4) Items 8 and 9 are now explained in a dedicated paragraph preceding Table 2 and in the Discussion; the explanation was placed in the text rather than as a table footnote to allow for a more thorough conceptual description. (5) We thank Dr. Landon for her positive comment on Table 6. (6) The MIPPE scale is now correctly uploaded as S1 File under Supporting Information.

---

## [Editor Report · Decision Letter 1]

5 Apr 2026

Development and validation of the MIPPE: a novel dyadic assessment tool for early parent-child interactions in clinical practice

PONE-D-25-61400R1

Dear Dr. Dufait,

We’re pleased to inform you that your manuscript has been judged scientifically suitable for publication and will be formally accepted for publication once it meets all outstanding technical requirements.

Kind regards,

Randall Waechter

Academic Editor

PLOS One

---

## [Editor Report · Acceptance letter]

PONE-D-25-61400R1

PLOS One

Dear Dr. Dufait,

I'm pleased to inform you that your manuscript has been deemed suitable for publication in PLOS One. Congratulations! Your manuscript is now being handed over to our production team.

Kind regards,

on behalf of

Dr. Randall Waechter

Academic Editor

PLOS One